# Detecting and Removing Adversarial Patches using Frequency Signatures

## Abstract

Computer vision systems deployed in safety-critical applications have proven to be susceptible to adversarial patches. The patches can cause catastrophic outcomes within autonomous driving scenarios. Existing defense techniques learn discriminative patch features or trigger patterns, which leave the defenses vulnerable to unseen patch attacks. In this paper, we propose Corner Cutter, a defense against adversarial patches that is robust to unseen patches and adaptive attacks. The framework is based on the insight that the construction process of adversarial patches leaves an attack signature in the frequency domain. The signature can be detected in different adversarial patches, including the LaVAN patch, the adversarial patch, the naturalistic patch, and a projected gradient descent-based patch. The framework neutralizes identified patches by isolating the high frequency signals and removing the corresponding pixels in the image domain. Corner Cutter is able to achieve an 11% increase in adversarial accuracy for the image classification task and an 8% increase in mean average precision on the Naturalistic patch over other defenses. The evaluations also demonstrate that the framework is robust to unseen patches and adaptive attacks.

## 1 Introduction

Object detection algorithms are pivotal to safety-critical domains such as autonomous vehicles. Nevertheless, the systems are vulnerable to adversarial attacks that manipulate sensor input data to cause incorrect output predictions. In the context of vision-based systems, these attacks first manifested as pixel adversarial examples (Szegedy et al., 2014), which are norm bounded, imperceptible perturbations. The adversarial examples evolved into adversarial patch attacks (Brown et al., 2018; Yang et al., 2020), which are more practical to realize in the real world (Xu et al., 2020). Adversarial patches have no norm bound but are instead constrained within a specified spatial region.

A number of defenses have been proposed to safe-guard computer vision models against adversarial patches. These defenses detect and neutralize adversarial patches based on image features (Das et al., 2017; Naseer et al., 2018), image masking (Levine & Feizi, 2020; McCoyd et al., 2020; Xiang et al., 2022a), and high intensity neurons and gradients (Han et al., 2021; Xiang et al., 2021). However, these defenses are often limited to the simple image classification scenario and cannot easily handle multiple objects or patches. More recent defenses predict the patch location(s) using a neural network (Liu et al., 2022; Tarchoun et al., 2023). This is achieved by training a neural network, or an auto encoder, to detect sample adversarial patches. This approach works well against patches that have previously been seen by the defense, but the defensive capabilities are diminished for unseen patches. This was the motivation behind the introduction of the naturalistic patch (Hu et al., 2021), which breaks the Segment and Complete defense in Liu et al. (2022). Instead of relying on easily modifiable patch features, it would be better for defenses to rely on an innate property of patches themselves so that even unseen patches will not be able to break the defense.

Patches are generated in a variety of ways, but all patch generation processes rely on a form of iterative noising in the patch region (Karmon et al., 2018; Liu et al., 2019). This results in the patch often being visibly different from the rest of the image in terms of color and content (Tarchoun et al., 2023). The visible differences between the image and the patch translate into additional differences when viewing images as signals in the frequency domain. The process of repeatedly adding unnatural noise to the patch modifies the magnitudes of frequencies across all frequency

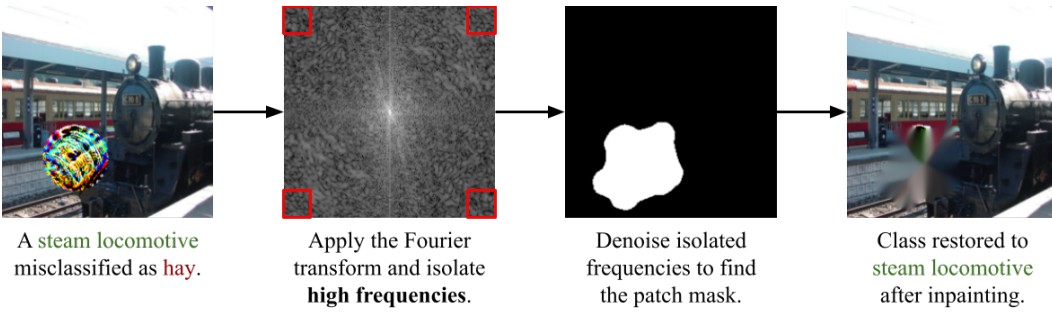

A steam locomotive misclassified as hay.

Apply the Fourier transform and isolate **high frequencies**.

Denoise isolated frequencies to find the patch mask.

Class restored to steam locomotive after inpainting.

Figure 1: An overview of the Corner Cutter framework.

bands, but the effects are most prominent in the higher frequencies. Those observations were used to develop a defense against adversarial examples by zeroing out the high frequencies (Zhang et al., 2019). However, the defense was broken by attacks from the competition hosted by Alibaba Group (2019). In this paper, we revisit frequency based defenses for adversarial patches. This is motivated by the radical difference between adversarial examples and adversarial patches. While it has been shown that a *spatially unconstrained* patch can modify specific portions of the frequency spectrum, it is not evident that the same can be achieved using a *spatially constrained* attack. Furthermore, we observe that patch attacks generate a recognizable frequency signature regardless of the patches starting state, whether it be image pixels, uniform color, or noise. This consistent signature is left in the frequency domain by the patch generation process and is present even for patches that use a starting point without high frequency components.

In this paper, we propose Corner Cutter, a defense against adversarial patches that exploits their high frequency signature to locate and remove them. An overview of the framework is presented in Figure 1. The main contributions of the paper are as follows:

- We show that adversarial patches tend to reside in higher frequency signals due to the patch generation process. The constructed patches contain high frequency signals no matter the patch type or the starting pixel values of the patch.
- We propose Corner Cutter, a novel defense against adversarial patches which uses the frequency domain to locate the patches. Previous defenses (Zhang et al., 2019) simply zeroed out high frequencies. Instead, Corner Cutter isolates high frequencies and transforms them back to the image domain to localize and eliminate the patch.
- We conduct experiments that show Corner Cutter is effective in both the image classification and object detection tasks. We also show that Corner Cutter is resistant to adaptive patches that minimize their frequency domain impact.

The remainder of this paper is organized as follows: preliminaries in Section 2, an analysis of patches in the frequency domain is provided in Section 3, the defense methodology is explained in Section 4, experimental results are given in Section 5, and the conclusion is in Section 6.

## 2 RELATED WORKS

### 2.1 PROBLEM FORMULATION

Adversarial patches are attacks on (i) image classifiers and (ii) object detectors that cause the models to produce incorrect outputs. Patches are generated from a variety of *starting points*, such as general adversarial network generated images (Hu et al., 2021) or textures from images in the target dataset (Yang et al., 2020). Nonetheless, all patch generation methods include a gradient descent process in which noise is iteratively added to the patch until an iteration or perturbation constraint is reached. An example of the noising process can be seen in the optimization problem used by the LaVAN patch $\hat{p}$ (Karmon et al., 2018):

$$\hat{p} = \operatorname{argmin}\mathbb{E}[logPr(y|f,n,x,l)], \tag{1}$$

where a bounded location $l$ on the target image $x$ is filled with noise $n$ which is updated at each gradient descent step such that the logit of the ground truth class $y$ is minimized when predicted

by the network $f$. Similarly, the Adversarial Patch $\hat{p}$ (Brown et al., 2018) is generated through the following optimization problem:

$$\hat{p} = \underset{p}{\text{argmax}}\mathbb{E}_{x\sim X, t\sim T, l\sim L}[logPr(\hat{y}|p, x, l, t)], \tag{2}$$

where some gray patch $p$ is treated as weights to be optimized during a gradient descent process in which the patch is applied to some set of images $X$ across different locations $L$ with a variety of transforms $T$ such that the probability of the target class $\hat{y}$ is maximized. The Naturalistic patch (Hu et al., 2021) and projected gradient descent patches are constructed using a similar process.

## 2.2 ADVERSARIAL DEFENSES

The most straightforward defense against adversarial patches involves iteratively masking different portions of an image to discover the ground truth. PatchCleanser (Xiang et al., 2022a) is a defense for image classification networks that employs this approach by using a double round masking procedure where the target image is masked in a grid-like fashion to search for any outliers in the network output. Similarly, ObjectSeeker (Xiang et al., 2022b) defends object detection networks by progressively masking rows and columns, aggregating all of the resulting bounding boxes, and pruning hallucinations. Masking defenses work well when presented with one patch, but have not been shown to be effective against multiple patches in one image.

Other popular defense methods search for high intensity network gradients or neurons so they can be masked or used as a starting point to trace back to the patch location. Patchguard (Xiang et al., 2021) uses networks with small receptive fields to avoid the effects of patches as well as a process that suppresses high intensity neurons within the model. Scalecert (Han et al., 2021) arbitrarily chooses one of the beginning layers of a network and traces the top neurons from that layer to the input to determine whether an image has been attacked. These defenses work well on image classification tasks, but have not shown effectiveness in object detection scenarios.

Other recent defenses include Jedi (Tarchoun et al., 2023), which defends both image classification and object detection networks by sliding a window across the target image to determine the locations with the highest entropy which are then inpainted. Jedi also uses an autoencoder to fill out the predicted patch location mask. Segment and Complete (Liu et al., 2022) uses a neural network to predict the patch location and a masking-based approach to fill in the patch shape. These defenses rely on networks trained with attacked image data, making them vulnerable to unseen patches.

A defense based on repeatedly cycling an image through spatial and frequency representations using the discrete cosine transform was proposed in Bafna et al. (2018). A defense against adversarial examples using frequency representations was proposed in Zhang et al. (2019). The defense zeroed out the high frequency signals in the Fourier domain. The defense can be broken using attacks from Alibaba Group (2019). This stems from the fact that *adversarial examples* are only norm bounded and not spatially constrained. We hypothesize and show that a similar adaptive attack cannot break the proposed defense due to the spatially bounded nature of patches.

## 2.3 FOURIER ANALYSIS

Images are two-dimensional signals and can be analyzed in the frequency domain using the two-dimensional (2D) discrete cosine transform or Fourier transform. The Fourier transform is widely used in a number of other domains including radar, heat transfer, and compression. The Fourier transform shows the constituent frequencies of a given signal in the form of sine and cosine waves. A transformed two-dimensional signal $G(f)$ can be represented through the following equation:

$$G(f) = \iint_{-\infty}^{\infty} g(x, y)e^{-j2\pi(k_x \times x + k_y \times y)}dx \times dy, \tag{3}$$

where $g(x, y)$ is the original signal, $j$ is $\sqrt{-1}$, $k_x$ and $k_y$ are frequencies of the signal in the $x$- and $y$-directions, and $x$ and $y$ are the time for one cycle, which corresponds to pixel coordinates when using images, in the respective directions. Peaks in the transformed signal represent the existence of the frequency corresponding to the peak's location in the original signal. For images, peaks in the frequency domain represent strong edges or other strong differences between image pixels. We use the Fast Fast Fourier Transform (Brigham & Morrow, 1967) to convert images from the spatial domain to the frequency domain in this paper.

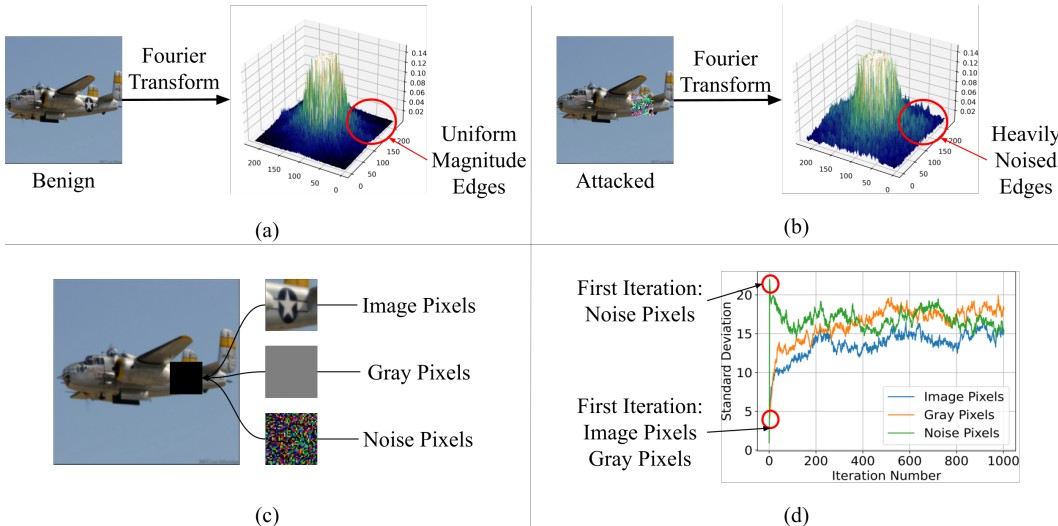

Figure 2: The frequency spectrum of a benign and attacked image is shown in (a) and (b), respectively. The standard deviation in the corners of the spectrum is low and high for the benign and attacked images respectively. Patches with different starting points are shown in (c). Regardless of the starting point, the patch generation process introduces noise with high standard deviation (frequency signature) in the corners of the frequency spectrum as seen in (d).

## 3 FREQUENCY SIGNATURES OF ADVERSARIAL PATCHES

In this section, we describe the frequency signature present in images subject to a patch attack. We also detail why images with adversarial patches will always contain a frequency signature. The concepts are outlined using the example in Figure 2.

**Frequency Signature of Adversarial Patches:** The frequency spectrum of a benign image obtained using the Fourier transform is shown in Figure 2(a). The values closer to the center represent lower frequencies while signals closer to the corners represent higher frequencies. The lower frequencies have high magnitudes since that is where the core of the image information resides. The higher frequencies have a small, uniform magnitude and approach zero towards the corners. The frequency spectrum of an image with an adversarial attack is shown in Figure 2(b). The entire frequency domain shows peaks and fluctuations throughout all the frequency bands. However, the most obvious distortions are located in the high frequency spectrum. The signal went from a uniform, low magnitude to noisy values with high variance. Moreover, there is little overlap between patches and the rest of the image in high frequencies, making the *corners* of the frequency spectrum the ideal location to detect the signature of an attack. In this paper, we define the standard deviation of a $k$-by-$k$ square in the corner of the frequency spectrum to be its *frequency signature*.

**Patch Generation Increases the Frequency Signature:** It is not surprising that a patch starting from (high frequency) random noise will result in a patch with a noticeable frequency signature. Next, we analyze the presence of the frequency signature for different patch starting points. Patches can start from the original image pixels, naturalistic patch, uniform color, or random noise, which is shown in Figure 2(c). For each of these starting points, we show the frequency signature (standard deviation in the corners) as a function of the iterations in the LaVAN patch generation process, which is shown in Figure 2(d). As expected, the different starting points have frequency signatures of different magnitude. The patch starting from the image pixels has the smallest signature and the random noise has the largest signature. Next, the patch is iteratively created by updating the patch using steepest gradient decent. It can be observed that regardless of the starting point, all of the final patches have a characteristically large frequency signature. We show that similar trends hold for the other patch generation processes in the appendix.

**Overview of the Corner Cutter Approach:** A straightforward approach to try and neutralize patches would be to zero out the corners with the frequency signatures. However, we find that this technique does not consistently neutralize the patch. Instead, the Corner Cutter framework converts the high frequency corners into the image domain to localize the patch.

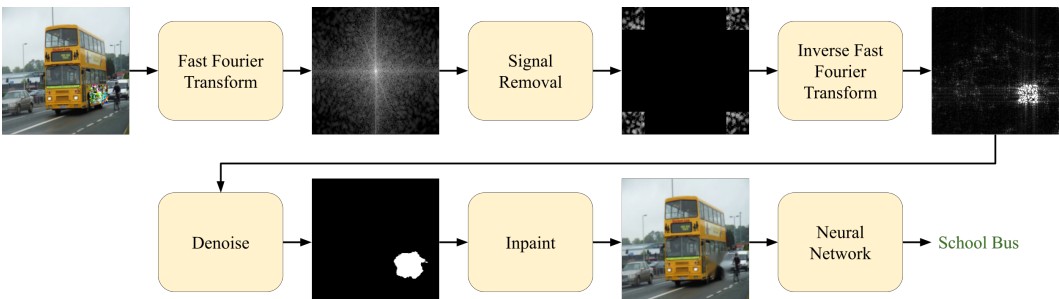

Figure 3: An attacked image is transformed into the frequency domain where all signals are removed except for those in the corners. The signals are then transformed back into an image that contains only the high frequency components of the original image. This high frequency image is denoised to get a mask that points to the location of the patch. Finally, the image is inpainted according to the mask and passed to a network for final predictions.

# 4 METHODOLOGY

In this section, we describe the methodology of the Corner Cutter framework. Corner Cutter can be split into three main steps: high frequency isolation, patch localization, and patch removal. The frequency isolation step transforms the image to the frequency domain and removes everything but the highest frequencies. The image is then transformed back to the image domain where the patch localization step denoises the image to find a mask for the patch location. Finally, the patch removal step repairs the image based on the patch mask for better network prediction. The framework flow is displayed in Figure 3.

## 4.1 ISOLATING HIGH FREQUENCIES

The isolation step starts with grayscaling and converting the target image to the frequency domain using the Fast Fourier Transform (Brigham & Morrow, 1967). Additionally, a phase shift is applied so that the low frequencies are at the center and the high frequencies are at the corners of the frequency domain. Next, everything but the corners of the image are removed. The removal process is outlined in the following equation:

$$f_c = \begin{cases} 0, & \text{if } x, y > s \text{ and } x, y < \text{length}(f) - s \\ f(x, y), & \text{otherwise} \end{cases}, \tag{4}$$

where the corner-only frequency domain $f_c$ is obtained by setting all pixels with x-coordinate $x$ and y-coordinate $y$ values outside of the corner squares of size $s$ to be 0 on the unmodified frequency domain $f$. We empirically find a strong upperbound of $s = 25$ using a sensitivity analysis in the appendix.

## 4.2 PATCH LOCALIZATION

After the frequency domain is cut, it is transformed back to the image domain using the inverse Fast Fourier Transform, where the patch localization step begins. The cut image is first thresholded and binarized. The operation is seen in the following equation:

$$i_{tb} = \begin{cases} 1, & \text{if } i_c(x, y) > \tau \times \text{max\_val}(i_c) \\ 0, & \text{otherwise} \end{cases}, \tag{5}$$

where the thresholded and binarized cut image $i_{tb}$ is created by setting any pixel $i_c(x, y)$ greater than some percentage $\tau$ of the cut image's $i_c$ maximum pixel value $max\_val$ to be 1 and setting all other pixels to be 0. This filters out most of the pixels that are not on the patch. Blurring and Otsu thresholding (Otsu, 1979) are then applied to generate a mask for the patch location. Additional analysis is provided in the appendix for the choice of threshold percentage $\tau$ and blurring kernel size. The denoising process is displayed in Figure 4.



Figure 4: After transforming the remaining high frequency signals into the image domain, binary thresholding, blurring, and Otsu thresholding are applied to the image to remove non-patch locations. The denoised image is used as the patch mask.

### 4.3 PATCH REMOVAL

Now that the patch location has been found, the patch can be removed. The patch location as referenced by the patch mask is either blacked out or inpainted. Blacking out the patch involves setting pixels in the mask to 0. For inpainting, there are many possible algorithms and models available for use. The specific inpainting approach used is less important than the impact of using inpainting over blacking out the pixels. We use the algorithm from Bertalmio et al. (2001) since it only incurs a small runtime overhead. After blacking out or inpainting the patch location, the restored image is passed to the model for a restored prediction.

## 5 EXPERIMENTAL RESULTS

In this section, we evaluate the Corner Cutter framework. The experiments were performed on a desktop and two servers. The desktop contains an NVIDIA A4000 with 12 CPU cores. Each server contains 4 NVIDIA A40s and 64 CPU cores. We evaluate Corner Cutter on multiple datasets, against different patches, and compare with other defenses. We use the Adversarial Patch (Brown et al., 2018) and Local and Visible Adversarial Noise (LaVAN) (Karmon et al., 2018) on the Imagenet (Deng et al., 2009) dataset, a projected gradient descent-based (PGD) patch (Madry et al., 2017) on the Xview dataset (Lam et al., 2018), and the Naturalistic Patch (Hu et al., 2021) on the INRIA Person dataset (Dalal & Triggs, 2005). We compare against Local Gradient Smoothing (Naseer et al., 2018), JPEG Compression (Das et al., 2017), and the defense outlined in Zhang et al. (2019), which we refer to as Remove Freqs, on both the image classification and object detection tasks. We also compare against PatchCleanser (Xiang et al., 2022a) on image classification and Segment and Complete (Liu et al., 2022) on object detection. For classifying images on Imagenet, we use the PyTorch (Paszke et al., 2019) implementation of Resnet50 (He et al., 2015), for detecting objects on Xview we use the Armory (TwoSixLabs, 2023) implementation of Faster RCNN (Ren et al., 2015), and for INRIA Person we use the Darknet (Redmon, 2013–2016) implementation of Yolov4-tiny (Wang et al., 2021). We use the Adversarial Patch and JPEG Compression implementation from the Adversarial Robustness Toolbox (Nicolae et al., 2018). The LaVAN and PatchCleanser implementations come from the PatchCleanser repository (InspireGroup, 2022). The PGD patch implementation is from the Armory. The Naturalistic patch implementation is from the original author's repository (Kung, 2021). The Segment and Complete implementation comes from the author's repository (Liu, 2021). The Local Gradient Smoothing implementation comes from the repository in Anonymous (2020). We use a personal implementation of the Remove Freqs defense.

The evaluations are broken into two subsections: an overall comparison of Corner Cutter on (i) image classification and (ii) object detection is provided in Section 5.1. The robustness of Corner Cutter against adaptive patches is analyzed in Section 5.2.

### 5.1 EVALUATION OF CORNER CUTTER

In this section, we evaluate the effectiveness of Corner Cutter for (i) image classification and (ii) object detection. We evaluate the performance of Corner Cutter for the image classification scenario in Table 1. The values in the table represent the accuracy on benign and attacked (adv) images. For each patch, we evaluated the defenses on the first 2000 Imagenet images that were correctly classified by Resnet50 when benign and misclassified when attacked. We used patches of size 1%, 2%, and 3% of image pixels for the LaVAN patch and 10%, 15%, and 20% of image pixels for the Adversarial Patch.

Table 1: Defense results against image classification patches using classification accuracy.

| Patch Type | Local and Visible Adversarial Noise | | | | | | Adversarial Patch | | | | | |
|---|---|---|---|---|---|---|---|---|---|---|---|---|
| Patch Size | 1% pixels | | 2% pixels | | 3% pixels | | 10% pixels | | 15% pixels | | 20% pixels | |
| Image Type | benign | adv | benign | adv | benign | adv | benign | adv | benign | adv | benign | adv |
| JPEG (Das et al., 2017) | 79.1 | 60.6 | 83.5 | 60.1 | 85.0 | 56.3 | 79.4 | 18.9 | 84.0 | 17.0 | 84.0 | 14.0 |
| LGS (Naseer et al., 2018) | 92.1 | 81.6 | 93.6 | 81.9 | 94.0 | 81.6 | 92.0 | 59.7 | 92.0 | 58.1 | 93.3 | 50.8 |
| PatchCleanser (Xiang et al., 2022a) | 91.9 | 35.8 | 92.6 | 35.0 | 92.8 | 33.5 | 87.0 | 30.8 | 83.5 | 28.5 | 75.7 | 34.4 |
| Remove Freqs (Zhang et al., 2019) | 91.5 | 53.1 | 93.4 | 47.5 | 93.9 | 37.8 | 88.5 | 24.8 | 90.7 | 23.6 | 91.0 | 17.7 |
| Corner Cutter (ours) | **95.5** | **94.7** | **94.9** | **94.5** | **94.4** | **92.7** | **98.7** | **70.9** | **98.3** | **68.7** | **98.0** | **60.8** |

It can be observed that Corner Cutter achieves an average of 96% benign accuracy and an $80\%$ average adversarial accuracy in Table 1. The average recovered adversarial accuracy is $42\%$, $11\%$, $47\%$, and $46\%$ higher than JPEG Compression (JPEG), Local Gradient Smoothing (LGS), PatchCleanser, and Remove Freqs. This stems from the defenses attempting to negate the patches without ever locating them, making it more difficult to successfully recover the correct classes. When comparing Corner Cutter with Remove Freqs, it can be observed that it is advantageous to locate the patch using high frequencies instead of simply zeroing them out. Moreover, all the defenses demonstrate a similar performance degradation in adversarial accuracy when comparing the two patches. This occurs because the Adversarial Patches are large. Even if a defense is able to perfectly locate the patch, there is still a large chunk of missing image information after patch removal, which makes it difficult to restore the original image label.

Table 2: Defense results against object detection patches using mean average precision (mAP).

| Patch Type | Project Gradient Descent Patch | | | | Naturalistic Patch | | | |
|---|---|---|---|---|---|---|---|---|
| Patch Size | Benign | 50x50 | 75x75 | 100x100 | Benign | 20% object | 25% object | 30% object |
| Undefended | 27.8 | 7.7 | 5.5 | 4.2 | 93.1 | 24.4 | 15.1 | 8.4 |
| JPEG (Das et al., 2017) | 23.7 | 10.9 | 9.0 | 6.9 | 88.2 | 17.7 | 6.6 | 5.7 |
| LGS (Naseer et al., 2018) | 23.0 | 7.0 | 5.7 | 3.6 | 91.8 | 35.4 | 14.3 | 10.2 |
| SAC (Liu et al., 2022) | **27.7** | **21.9** | 15.5 | 9.5 | **93.1** | 31.0 | 12.9 | 7.2 |
| Remove Freqs (Zhang et al., 2019) | 27.1 | 8.9 | 7.5 | 4.1 | 92.6 | 24.4 | 9.8 | 8.2 |
| Corner Cutter (ours) | 26.6 | 15.8 | **16.6** | **14.1** | 89.7 | **47.1** | **22.1** | **11.5** |

The object detection scenario is evaluated in Table 2. The table reports the mean average precision (mAP) with respect to the patch size. The undefended model results are provided for reference. For the PGD patch, we used the first 2000 images from the Xview dataset. For the Naturalistic patch we applied 25 unique patches to each sample of the 288 image INRIA Person test set resulting in 7200 unique defense scenarios and averaged the resulting values. The PGD patch used patches that were of 50-by-50, 75-by-75, and 100-by-100 pixels. The Naturalistic patches were resized to be $20\%$, $25\%$, and $30\%$ of each person detected in an image, with potentially multiple patch occurrences.

It can be observed that Corner Cutter and Segment and Complete (SAC) produce the best results across both patches. The performance for the PGD patch on the Xview dataset are similar for both defenses. Corner Cutter and SAC respectively achieve average mAPs of $15.5\%$ and $15.6\%$ across the different patch sizes. However, the performance of SAC is substantially worse than Corner Cutter on the INRIA dataset when evaluated using the Naturalistic patch. Corner Cutter and Segment and Complete achieved average mAPs of $26.9\%$ and $17.0\%$, respectively. The worse performance of SAC for the naturalistic patch is easy to understand. The SAC framework was trained on the PGD patch and has learned its triggers. However, the Naturalistic patch is an unseen patch and therefore

Local and Visible
Adversarial Noise  Adversarial Patch  Naturalistic Patch

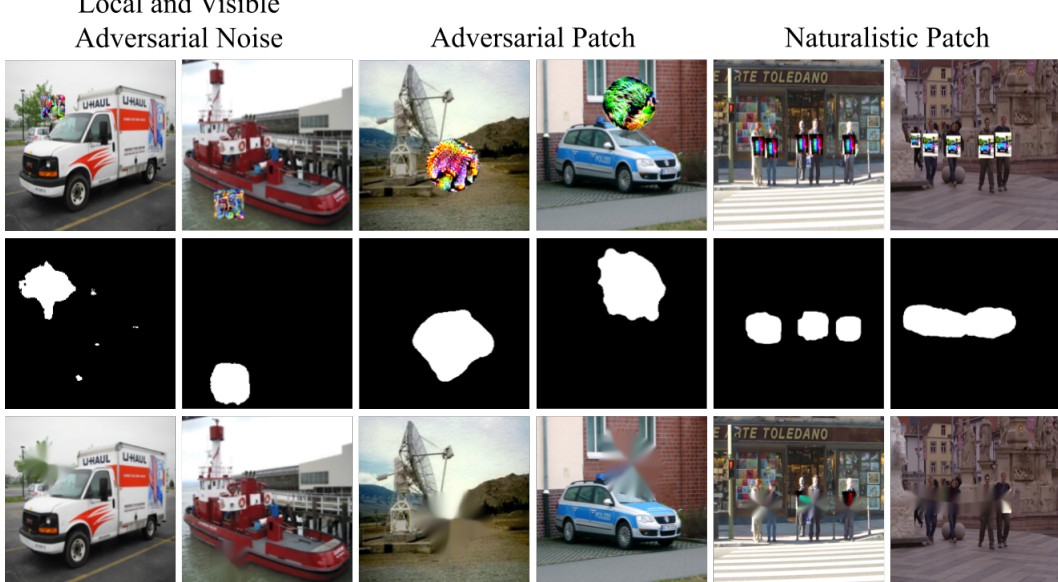

Figure 5: Examples of Corner Cutter defending against attacks.

the performance is worse. In contrast, Corner Cutter does not rely on training a neural network to recognize patch features. Instead, it neutralizes patches based on the frequency signature from the patch generation process. While we were unable to directly compare with Jedi (Tarchoun et al., 2023), they reported results of 64% attacked accuracy compared to our 67% on the Adversarial Patch and 28% recovered mAP compared to our 47% on the Naturalistic patch on the same datasets provided in our results.

It is interesting to note that using inpainting yielded better results on the classification patches while blacking out yielded better results on object detection patches. We posit that inpainting helps classification networks since patches are filled with pixels from the surrounding location, thus resulting in more values which point toward the original class, whereas blacking out removes the location from the network's view altogether, taking with it important class information. The opposite would be true for object detection networks, where inpainting can potentially skew object bounding box sizes or hallucinate objects, while blacking out removes the information, leading to less places available for prediction.

We include a selection of qualitative results for Corner Cutter in Figure 5. Each column shows a different image, and the rows from top to bottom show the attacked image, the patch mask found by Corner Cutter, and the final inpainted image.

## 5.2 Evaluation on Adaptive Attacks

In this section, we evaluate Corner Cutter on two different adaptive attacks. Corner Cutter identifies the patch location using the frequency signature in the high frequency spectrum. Intuitively, an adaptive attack against Corner Cutter should try to suppress the magnitude of high frequencies. We evaluate the robustness of Corner Cutter for both the (i) image classification and (ii) object detection scenarios.

We design an adaptive version of the LaVAN patch for image classification. We make the LaVAN patch adaptive by limiting the size of the noise introduced to all frequencies in the frequency domain at each step of gradient descent. If the change is larger than the threshold, we project the frequencies back into the feasible domain. We test over thresholds from 0.01 to 0.24 on a 0.01 increment. The results of Corner Cutter evaluated on an adaptive and non-adaptive LaVAN patch are shown in Table 3. The accuracies shown for each patch size for the adaptive patch are averaged across each of the allowed frequency perturbations. The attack success rate is also shown in the table. Reducing the size of the permissable changes in the frequency domain significantly harms the accuracy of Corner

Table 3: Evaluation of Corner Cutter on an adaptive LaVAN patch.

| Patch Size | Accuracy (%) on Local and Visible Adversarial Noise | | | Attack Sucess Rate (%) | | |
|---|---|---|---|---|---|---|
| | 1% image | 2% image | 3% image | 1% image | 2% image | 3% image |
| Non-adaptive | 94.7 | 94.5 | 92.7 | 83.0 | 95.9 | 99.4 |
| Adaptive | 39.2 | 39.7 | 39.4 | 25.1 | 38.4 | 48.2 |

Cutter, since it becomes more challanging to pick up the patch frequency signature. While accuracy drops from around 93% to around 39%, the defense is not completely broken. Many adaptive attacks are capable of reducing the accuracy into the single digits. Moreover, the additional constraints harm the patch generation capabilities, which is shown to the right in the table. The patch generation success rate is reduced from over 83% - 99% to the 25% - 48% range. Consequently, an attacker would have to make 3 - 4 attempts before a successful patch would be generated. Nevertheless, it is notoriously difficult to defend image classifiers on ImageNet since there are many highly related classes in the dataset. When a portion of the image is covered, it becomes immensely difficult to separate these similar classes from each other.

For the more important object detection scenario, we design an adaptive attack for the more challenging Naturalistic patch. The patch is made adaptive by adding in the average value of the 25-by-25 corner squares in the frequency domain as an additional loss term. Adding a loss term is the standard approach of implementing an adaptive attack. This mimics the approach in the image classification scenario, and makes the adaptive attack compatible with the patch generation process of the Naturalistic patch. The results of Corner Cutter evaluated on an adaptive and non-adaptive Naturalistic patch are shown in Table 4. The mAPs for the adaptive patch are averaged across the results of 5 patches applied to each of the 288 images in the INRIA Person test set totaling 1440 unique defense scenarios. The mAP for the two attacks is nearly the same. Trying to minimize the magnitudes of the high frequencies via a loss term does not generate a patch that can affect Corner Cutter. This demonstrates that Corner Cutter is robust to adaptive attacks against object detectors. This indicates that it is not possible to modify a *specific portion* of the frequency domain, while being *constrained* in the *spatial domain*. This explains why the adversarial examples were successful Alibaba Group (2019) against the frequency based defence in (Zhang et al., 2019), while our defence Corner Cutter proves to be robust to adaptive attacks.

Table 4: Evaluation of Corner Cutter on an adaptive Naturalistic patch.

| Patch Type | Naturalistic Patch | | |
|---|---|---|---|
| Patch Size | 20% object | 25% object | 30% object |
| Non-adaptive | 47.1 | 22.1 | 11.5 |
| Adaptive | 48.1 | 20.3 | 11.4 |

## 6 CONCLUSION

The iterative noising process used to generate patches leads to the introduction of noise in the frequency domain. This noise is most apparent in the higher frequencies and becomes readily apparent when compared with benign images. We propose Corner Cutter, a defense against classification and object detection patches that takes takes advantage of the high frequency signatures by transforming images to the frequency domain and isolating their high frequencies to locate patches. As all patches contain these high frequency signatures regardless of starting point, Corner Cutter is effective against unseen patches. We show that Corner Cutter outperforms other defenses in the image classification task and performs on par with or outperforms other defenses in the object detection task. We also show that Corner Cutter is resilient to adaptive patches that have been modified to minimize the high frequency signature left by patches.

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

## A  OVERVIEW

The appendix contains 6 additional sections. We list the hyperparameters used to generate each attack and the parameters used for each defense in Section B. We examine additional frequency signatures of more patches in Section C. We perform ablation studies on the value used for the binary threshold and the size of the kernel used in blurring in Section D. We qualitatively evaluate an optimal value for the size of the corners used in Corner Cutter in Section E. We evaluate Corner Cutter on one additional classification patch in Section F. Finally, we give extra examples of Corner Cutter defending against attacks across all of the patches used in the initial evaluation in Section 5.

## B  HYPERPARAMETERS USED IN EXPERIMENTS

In this section, we list all of the hyperparameters used by the attacks and defenses to obtain the results found in Tables 1 and 2. We start with attacks and then move on to defenses.

We used the LaVAN patch implementation in InspireGroup (2022). We used the parameters $epsilon = 1$, $steps = 500$, $step\_size = 0.05$, and $random\_start = True$. We used the Adversarial Patch implementation in Nicolae et al. (2018). We set all of the transform parameters to be 0 and use $batch\_size = 32$, $learning\_rate = 0.1$, and $max\_iter = 1000$. We used the PGD patch implementation in TwoSixLabs (2023). We used the parameters $batch\_size = 1$, $eps = 1$, $eps\_step = 0.004$, $max\_iter = 500$, $num\_random\_init = 0$, $random\_eps = False$, and $targeted = False$. We used the author's implementation of the Naturalistic patch in Kung (2021). We left all of the parameters in their default states.

We used the JPEG Compression implementation from Nicolae et al. (2018). We used the parameter $quality = 50$. We used the Local Gradient Smoothing implementation from Anonymous (2020). We used the parameters $threshold = 0.2$, $smooth\_factor = 2.3$, $block\_sizes = 15$, and $overlap = 5$. We used the author's implementation of PatchCleanser at InspireGroup (2022). We used the parameter $num\_mask = 6$. We used our own implementation of Zhang et al. (2019) and set $corner\_size = 75$. We used the author's implementation of Segment and Complete from Liu (2021) and use their default parameters. We explain our selection of Corner Cutter parameters in Sections D and E.

## C  ADDITIONAL FREQUENCY SIGNATURES IN THE FREQUENCY DOMAIN

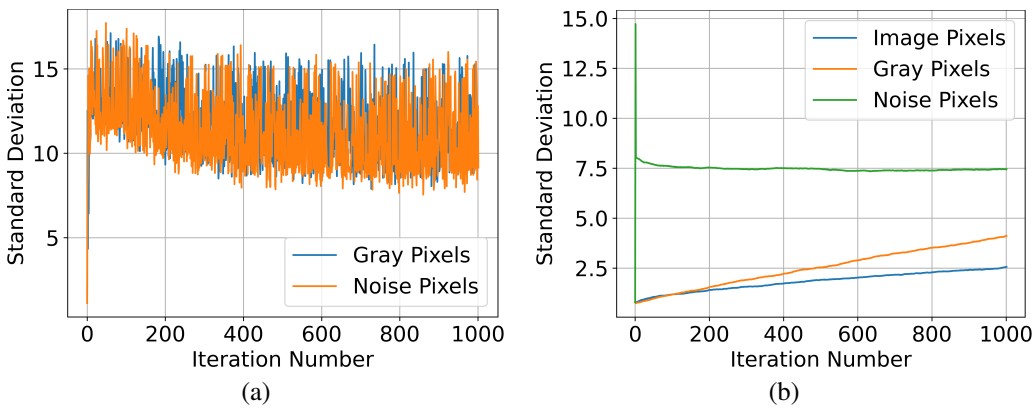

Figure 6: Frequencies of the (a) Adversarial Patch and (b) PGD patch with different starting points.

In this section, we analyze the frequency signatures of the Adversarial Patch and PGD patch with different starting points as a function of the iterations in the patch generation process. The frequency signature of the Adversarial Patch when starting from gray and noise pixels is shown in Figure 6(a). The Adversarial Patch is a universal patch and is therefore trained over a batch of images, so we do not include the frequency signature when starting from image pixels. There is no distinguishable difference between the beginning iteration's frequency signatures of the two starting points. The

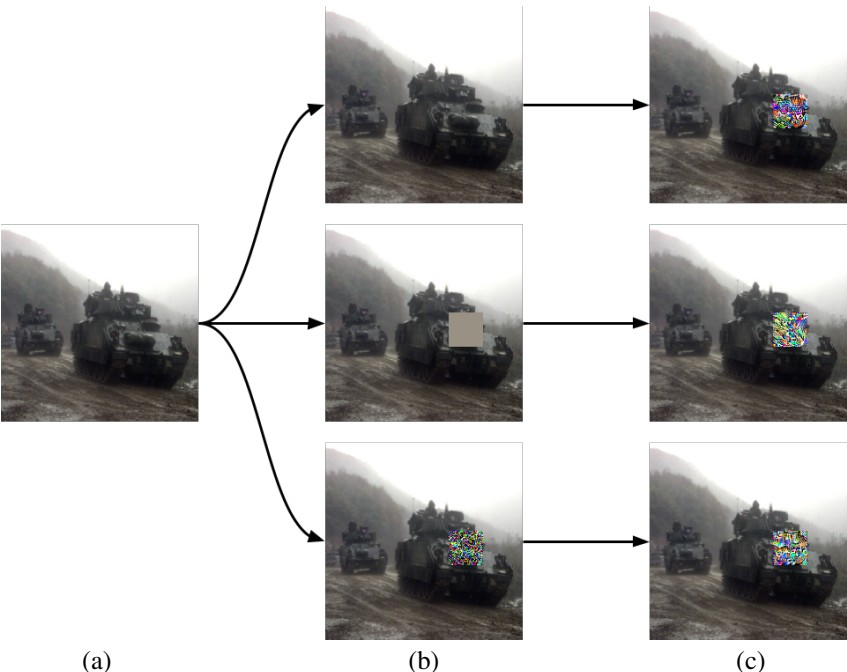

Figure 7: (a) The input image. (b) The initial patch state of, from top to bottom, image pixels, gray pixels, and noise pixels. (c) The final patch after generation.

Adversarial Patch approach treats the patch as weights to be optimized, as opposed to updating patch pixels solely based on loss like the LaVAN and PGD patches, so the patch values will take bigger steps at the beginning of optimization, resulting in beginning iterations that are similar to a noisy starting point. As expected, the frequency signatures spike quickly in the beginning iterations before somewhat converging.

The frequency signatures of the PGD patch when starting from image pixels, gray pixels, and noise pixels is shown in Figure 6(b). The noise starting point acts as expected, rapidly increasing in the first few iterations before decreasing in magnitude and converging. The image and gray starting points have frequency signatures that very gradually increase from their initial values, which is different than the rapid spikes seen in the Adversarial Patch and LaVAN patch. This can be explained by the $L_0$ constraint on our PGD patch. The three starting points do not converge to the same value within the allowed 1000 iterations, but they do all end with a higher frequency signature than their benign state.

We also qualitatively show that the patch will look similar no matter the starting state in Figure 7. For each of the starting states, the patch finishes as noise. The patches are not exactly the same, but are very similar and will have similar high frequency signatures.

## D CORNER CUTTER HYPERPARAMETER ABLATION STUDIES

In this section, we present ablation studies on two of Corner Cutter's hyperparameters: the binary threshold percentage $\tau$ in Figure 8 and the blurring kernel size in Figure 9. We used the 2% pixels LaVAN patch, 15% pixels Adversarial Patch, 75-by-75 pixels PGD patch, and 25% object Naturalistic patch. Each parameter step for the LaVAN patch and Adversarial Patch was evaluated on 1000 images. The parameter steps of the PGD patch were only evaluated on 100 images due to the significant time overhead for generating the patches. The parameter steps for the Naturalistic patch were each evaluated on 25 unique patches each applied to all 288 INRIA Person Dalal & Triggs (2005) test set images, resulting in a total of 7200 defense samples per parameter step.

The threshold percentage ablation study across the four patches is shown in Figure 8. The threshold percentage was varied from 0.0 to 1.0 on 0.05 increments. The blurring kernel size was fixed at

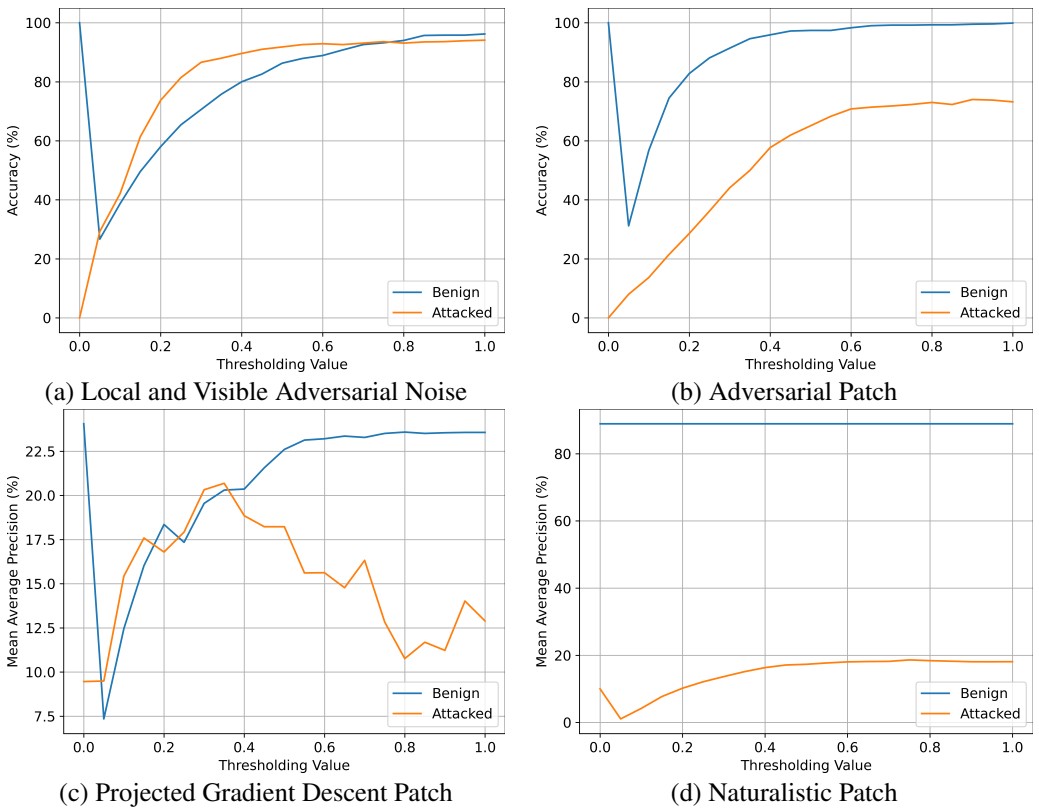

Figure 8: Corner Cutter's benign and attacked accuracy and mean average precision on the (a) LaVAN patch, (b) Adversarial Patch, (c) PGD patch, and (d) Naturalistic patch for different thresholding values.

$40$ for this experiment. The benign accuracy and mean average precision (mAP) across the patches quickly drops between $0.0$ and $0.05$, but steadily increases from there before converging at $0.8$ at the latest. Similarly, the restored accuracies for the LaVAN patch, Adversarial Patch, and Naturalistic patch plateau at $0.6$, $0.8$, and $0.6$, respectively. The PGD patch reaches a peak at $0.35$ before decreasing again. The latest convergence point for both benign and attacked images, excluding the PGD patch, is at $0.8$, so we use that value in all evaluations for Corner Cutter.

The blurring kernel size ablation study across the four patches is shown in Figure 9. The kernel size was varied from $1$ to $100$ on increments of $1$. The threshold percentage size was fixed at $0.8$ for this experiment. The benign accuracy and mAP dropped as the kernel size increased across each patch, so using a smaller kernel size is preferable for maximizing benign results. However, there is no pattern for attacked accuracies between the patches, making it difficult to pinpoint a singular best value to use. For those looking to use our work in their own evaluations without having to do this kind of analysis, a kernel size of $30$ would be best since it is optimal for the LaVAN patch and Naturalistic patch and near optimal for the adversarial patch. We used kernel sizes of the same size as the patch under evaluation since there was no consensus on the best kernel size to use.

## E    Isolation Size Qualitative Analysis

In this section, we qualitatively evaluate the images that come as a result of Corner Cutter isolating high frequencies with different corner side lengths $s$. Images with increasing $s$ are shown in Figure 10. As $s$ increases, the patch location becomes more clear and more precise. As discussed in Section 3, the patch is not only contained in the highest frequencies, so it makes sense that as more frequencies are isolated, the patch becomes clearer. The drawback of including more frequencies is that more image information is included with the patch information, leading to more noise when the

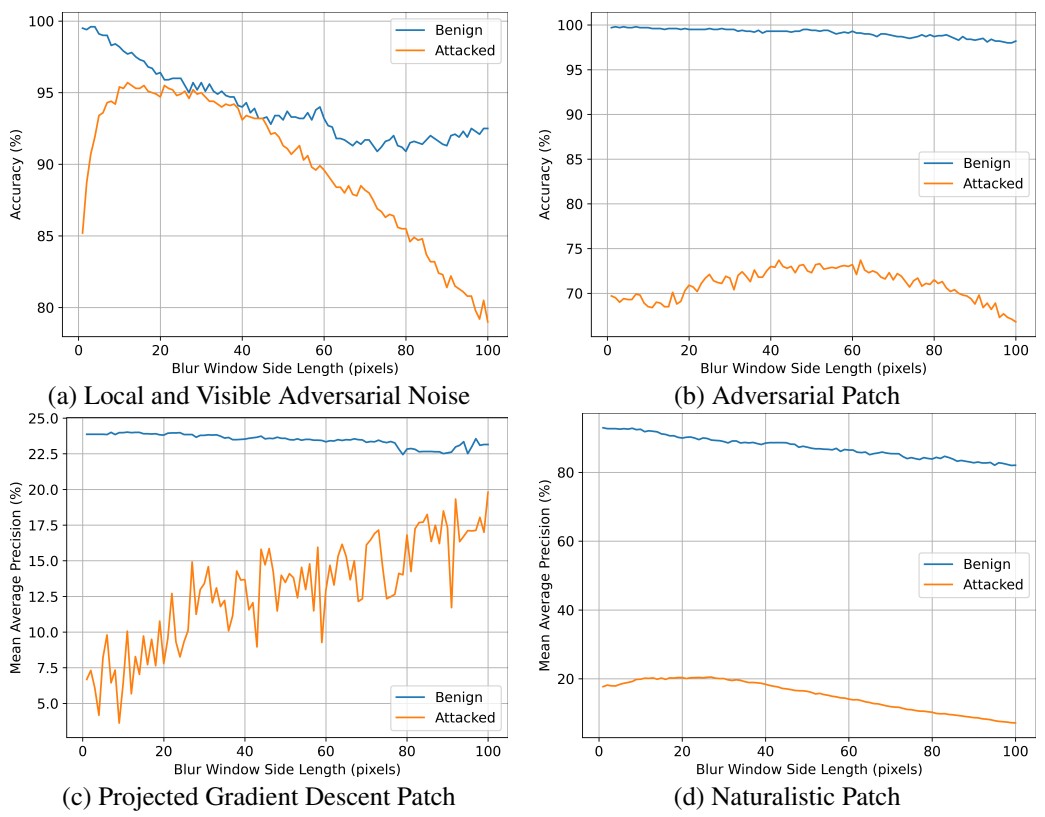

(a) Local and Visible Adversarial Noise

(b) Adversarial Patch

(c) Projected Gradient Descent Patch

(d) Naturalistic Patch

Figure 9: Corner Cutter's benign and attacked accuracy and mean average precision on the (a) LaVAN patch, (b) Adversarial Patch, (c) PGD patch, and (d) Naturalistic patch for different blurring kernel sizes.

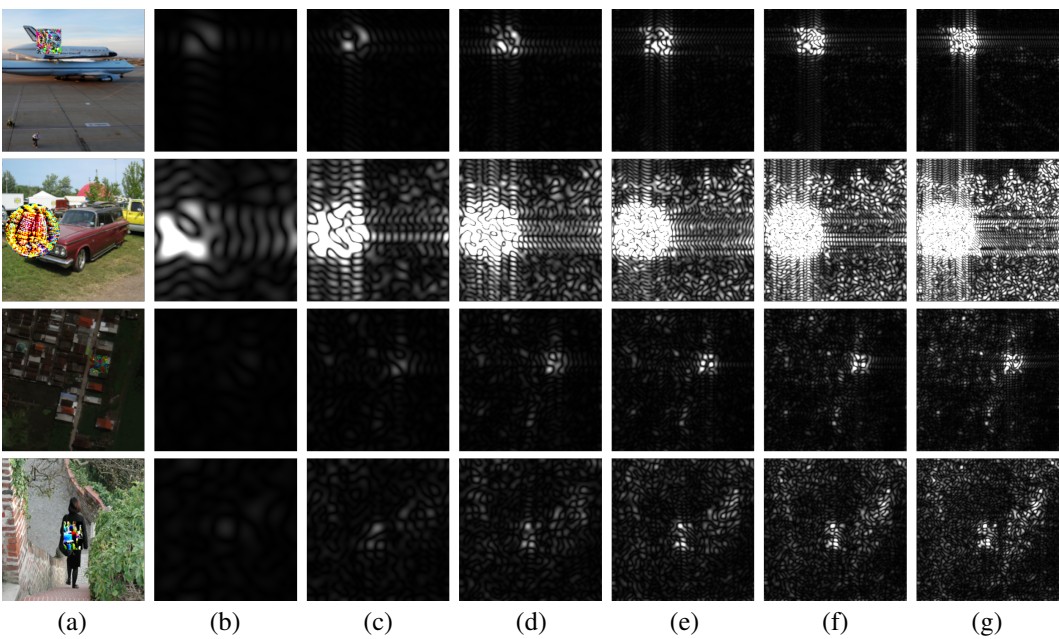

(a)  (b)  (c)  (d)  (e)  (f)  (g)

Figure 10: An (a) attacked image after Corner Cutter isolates high frequencies with corner side lengths of (b) 5, (c) 10, (d) 15, (e) 20, (f) 25, and (g) 30 pixels.

Table 5: Evaluation of Corner Cutter on the Texture-based Patch.

| Patch Type | | Texture-based Patch | | | | | |
|---|---|---|---|---|---|---|---|
| Patch Size | | 10% pixels | | 15% pixels | | 20% pixels | |
| Image Type | | benign | adv | benign | adv | benign | adv |
| JPEG (Das et al., 2017) | | 77.5 | 18.8 | 82.4 | 20.47 | 82.2 | 14.0 |
| LGS (Naseer et al., 2018) | | 89.6 | 26.0 | 93.0 | 19.7 | 92.0 | 15.4 |
| PatchCleanser (Xiang et al., 2022a) | | 83.9 | 14.6 | 81.2 | 21.4 | 73.3 | 26.9 |
| Remove Freqs (Zhang et al., 2019) | | 84.7 | 20.0 | 88.9 | 22.4 | 89.7 | 15.5 |
| Corner Cutter (ours) | | **98.4** | **56.0** | **98.6** | **51.9** | **97.9** | **47.8** |

cut frequency domain is transformed back to the image domain. For those wanting to use Corner Cutter without any additional calibration, the patch clearness tends to saturate between $s = 20$ and $s = 25$, so we recommend picking a value in that range. These are not always the optimal values, though, as it is easier for Corner Cutter to recover the Adversarial Patch image (second row of Figure 10) with $s = 5$ than $s = 25$ due to the over abundance of high magnitude noise. Ultimately, the optimal value will vary between patches, but $s = 20$ to $s = 25$ is a good baseline to use.

## F  ADDITIONAL PATCH EVALUATION

The results of JPEG Compression (Das et al., 2017), Local Gradient Smoothing (Naseer et al., 2018), PatchCleanser (Xiang et al., 2021), Remove Freqs (Zhang et al., 2019), and Corner Cutter evaluated on the Texture-based patch (Yang et al., 2020) are shown in Table 5. The values in the table represent benign and attacked (adv) classification accuracy. We generated patches of size 10%, 15%, and 20% of the image. For each patch size, we evaluated defenses on the first 2000 images that were both correctly classified when benign and were successfully misclassified to the target class when attacked.

Corner Cutter achieves an average benign accuracy of 98% and an average attacked accuracy of 52%, the highest of all the defenses. The next highest benign and attacked accuracies are 92% and 21% from Local Gradient Smoothing and PatchCleanser, respectively. The attacked accuracy for all defenses on the Texture-based patch are lower than those of the Adversarial Patch (Brown et al., 2018) and LaVAN patch (Karmon et al., 2018). The Texture-based patch uses a reinforcement learning algorithm to place the patch in an optimal location. This location is usually on top of, or at least intersecting with, the object that is the class of the image. This placement makes it difficult to recover the image class even if the patch location is correctly found, resulting in the overall lowered recovered accuracies.

## G  QUALITATIVE RESULTS

Examples of Corner Cutter defending against the LaVAN patch, Adversarial Patch, PGD patch, and Naturalistic patch are shown in Figure 11, Figure 12, Figure 13, and Figure 14, respectively. A benign image has the respective patch applied to it and becomes an attacked image. Once the attacked image is passed to Corner Cutter, it is transformed to the frequency domain using the Fourier transform. The high frequencies are isolated and then the cut frequency domain is transformed back to the image domain. The cut image is denoised to get a patch mask. Finally, the attacked image is inpainted based on the patch mask.

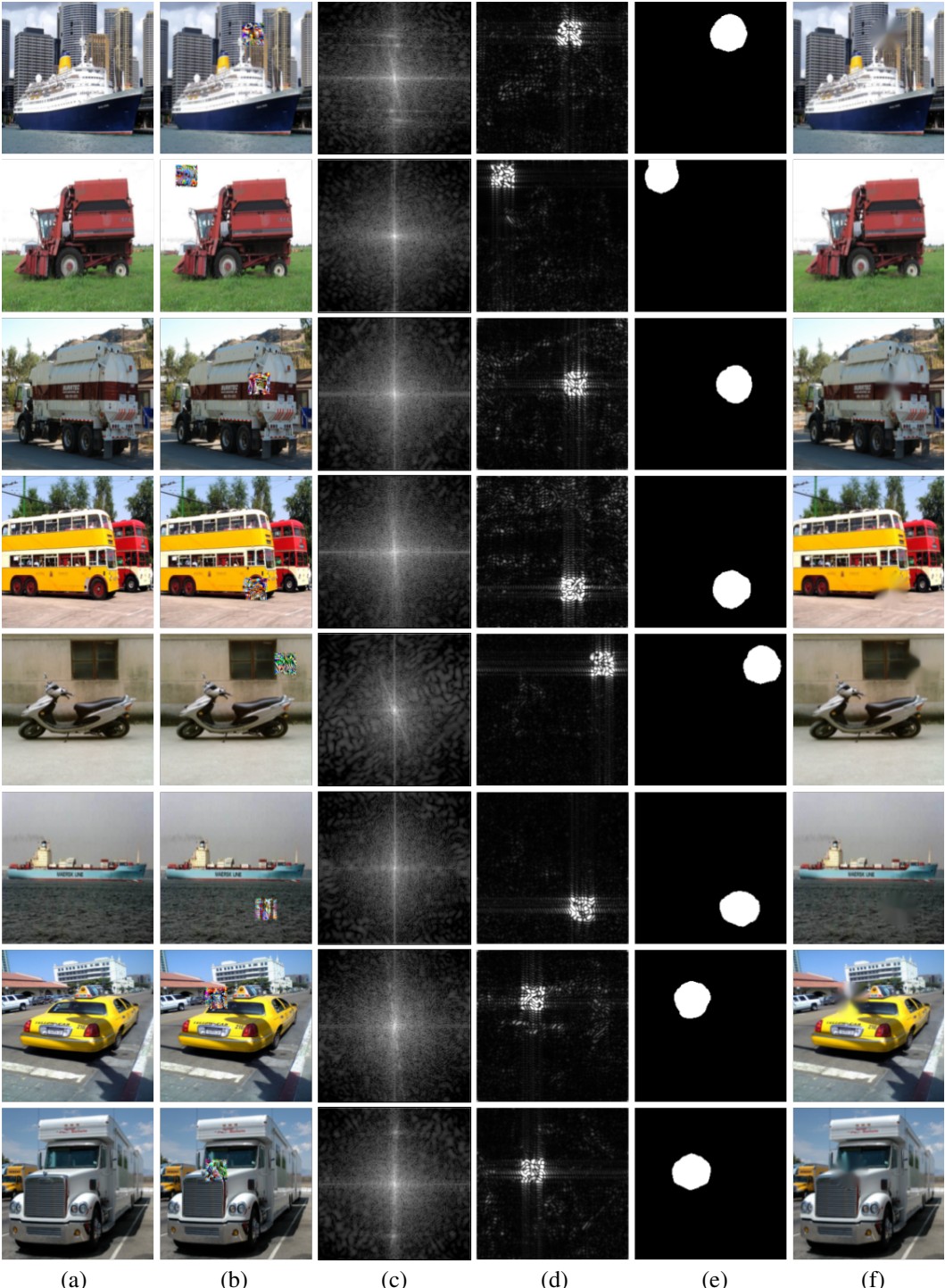

(a)  (b)  (c)  (d)  (e)  (f)

Figure 11: (a) A benign image. (b) A LaVAN patch applied to the image. (c) The frequency domain. (d) The cut image. (e) The patch mask after denoising. (f) The attacked image inpainted based on the patch mask.

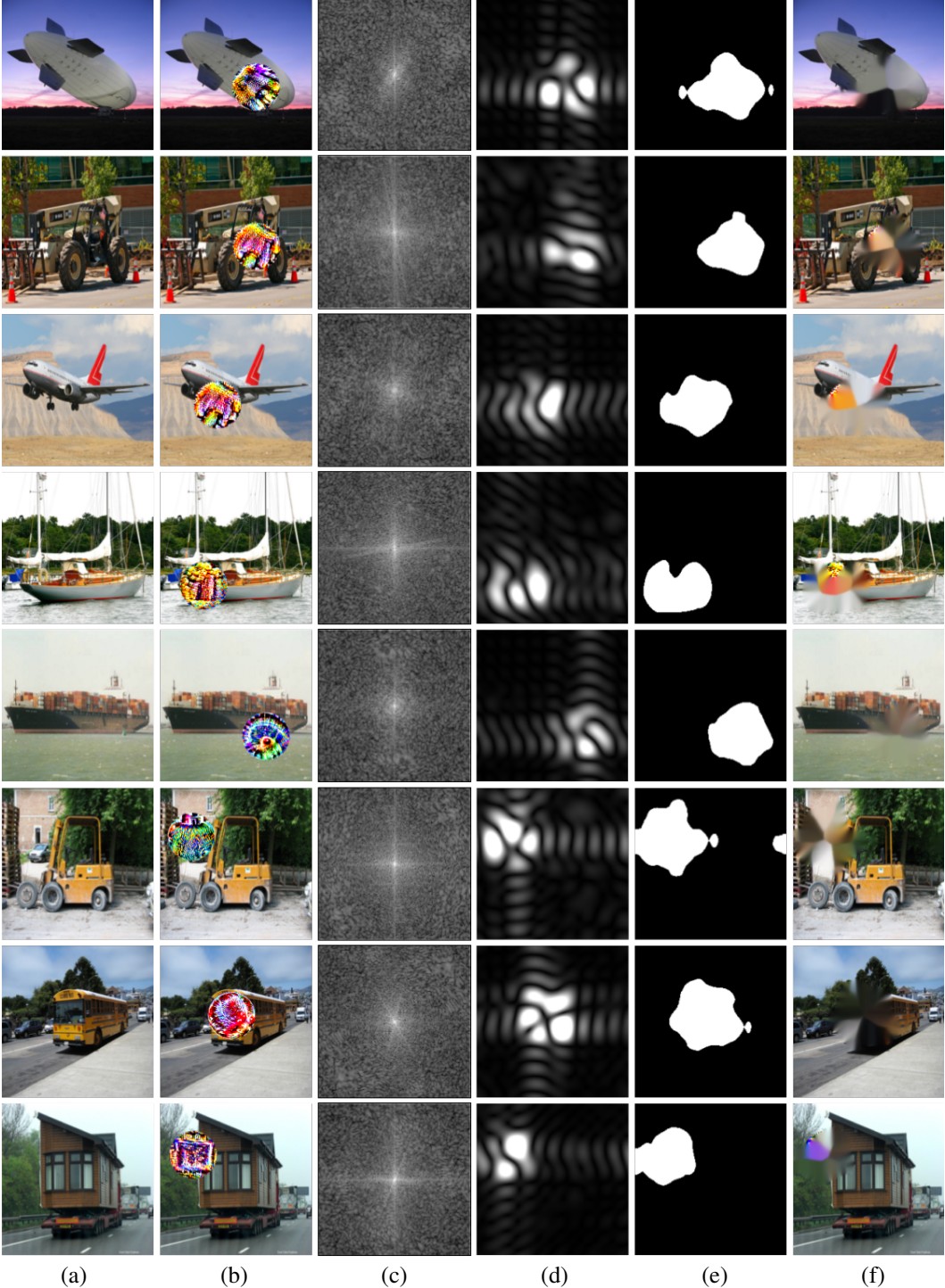

Figure 12: (a) A benign image. (b) An Adversarial Patch patch applied to the image. (c) The frequency domain. (d) The cut image. (e) The patch mask after denoising. (f) The attacked image inpainted based on the patch mask.

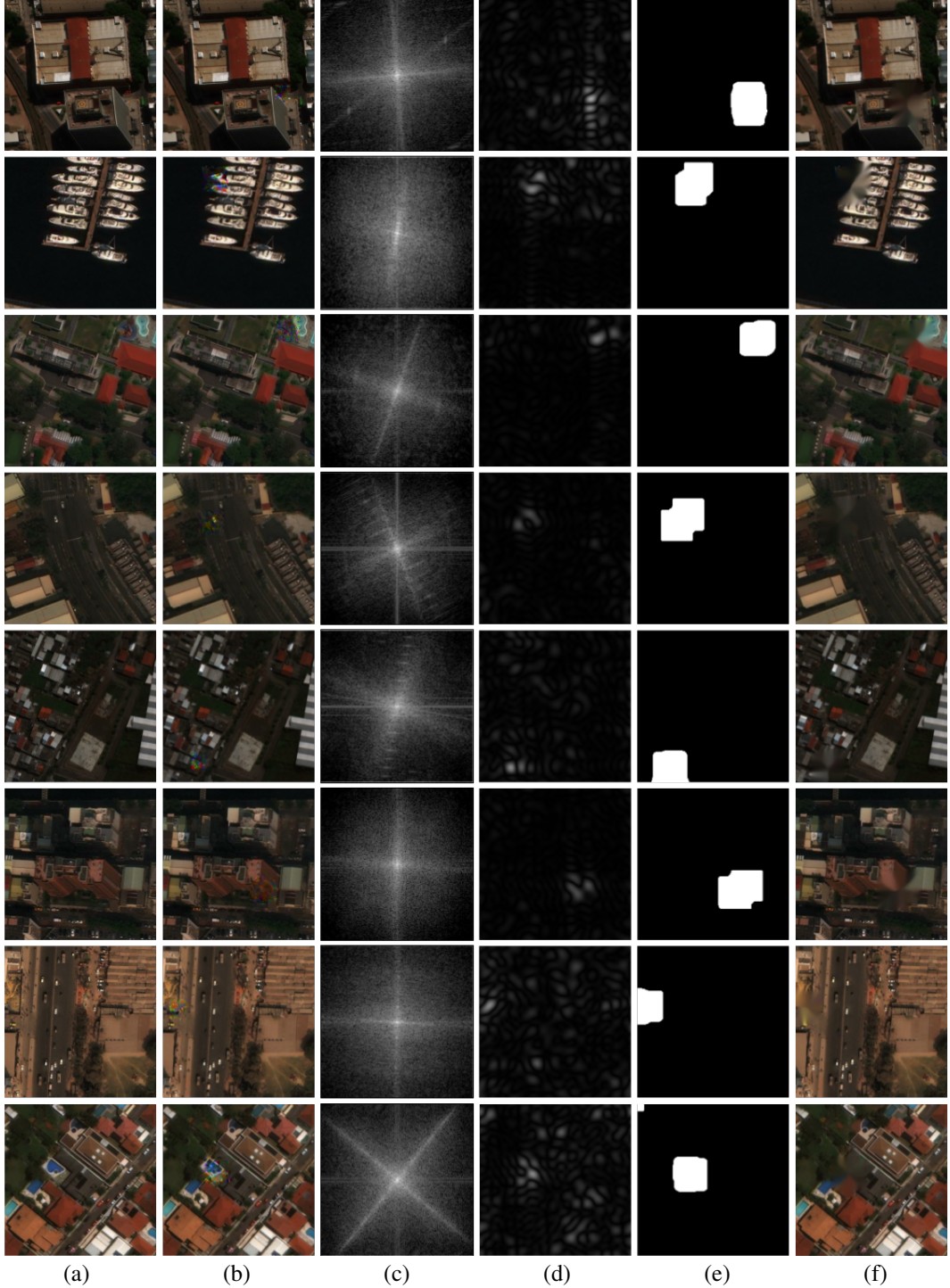

Figure 13: (a) A benign image. (b) A PGD patch applied to the image. (c) The frequency domain. (d) The cut image. (e) The patch mask after denoising. (f) The attacked image inpainted based on the patch mask.

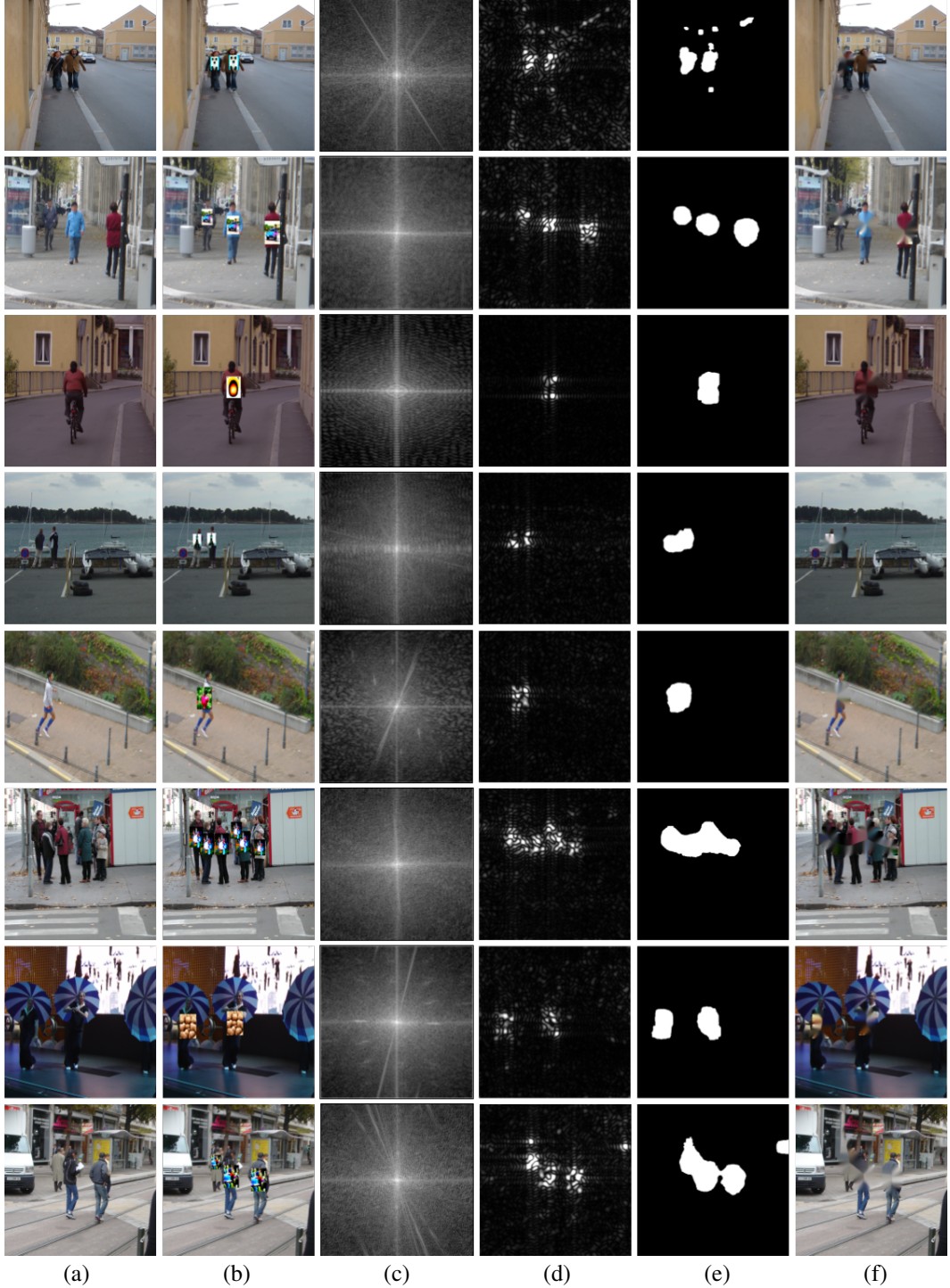

Figure 14: (a) A benign image. (b) A Naturalistic patch applied to the image. (c) The frequency domain. (d) The cut image. (e) The patch mask after denoising. (f) The attacked image inpainted based on the patch mask.

