# OpenReview forum: "Detecting and Removing Adversarial Patches using Frequency Signatures"
_ICLR.cc/2024/Conference — ICLR 2024 Conference Withdrawn Submission_

### Official Review · Reviewer_pCXn · 2023-10-29

**Soundness:** 3 good
**Presentation:** 2 fair
**Contribution:** 2 fair
**Rating:** 5
**Confidence:** 4

**Summary:**

This paper introduces Corner Cutter, a defense mechanism designed to counter adversarial patches. The foundation of this defense lies in the observation that the creation of adversarial patches introduces a distinctive attack signature in the frequency domain. This signature can be detected across various adversarial patches. Corner Cutter neutralizes these identified patches by isolating high-frequency signals and eliminating the corresponding pixels in the image domain. The experiment results show the effectiveness and robustness of the proposed defense.

**Strengths:**

1 The performance of Corner Cutter is good.
2 The evaluation involves the use of adaptive attacks.

**Weaknesses:**

1 The analysis in section 3 lacks depth, as it primarily offers a definition of the frequency signature without delving into a comprehensive theoretical or statistical examination of the distinctions among various attack patches.
2 The experimental section is deficient in an ablation study, which assesses the effectiveness of each individual component within the proposed defense.

**Questions:**

1 It is not clear whether Fig 2 (d) is for one image or multiple images.
2 The author asserts that the proposed defense employs the frequency domain to identify patches. What is the average precision achieved using this technique for various types of adversarial patches?
3 How does the performance change when only the locating and erasing components are applied? Since inpainting alone can alter the image content, it's plausible that random inpainting might effectively deteriorate the attack performance.
4 Have you taken into account the possibility of comparing your method with augmentation techniques such as cutout or cutmix, which intuitively aid in countering adversarial attacks stemming from adversarial patches?

---

### Official Review · Reviewer_1H9u · 2023-10-29

**Soundness:** 2 fair
**Presentation:** 2 fair
**Contribution:** 2 fair
**Rating:** 3
**Confidence:** 5

**Summary:**

The paper proposes a frequency-domain defense against adversarial patch attacks. The proposed algorithm, Corner Cutter, transforms the image into the frequency domain and leverages high-frequency components to locate the patch region for patch removal. The defense is claimed to be effective for both image classification and objection and robust against adaptive attacks.

**Strengths:**

1. studied an interesting and important problem of adversarial patch defenses
2. the idea is simple and intuitive

**Weaknesses:**

Overall, the experiments do not convince me that this is an effective defense.

A. The evaluation setup and the results of Table 1 look odd.
1. The robust accuracy of PatchCleanser is much lower than the numbers reported in this original paper. For example, the robust accuracy for a 2%-pixel patch for ResNet50-based PatchCleanser is 53% for the *entire* ImageNet dataset in its original paper. However, even though this paper reports performance for *2000 correctly classified* images, the reported robust accuracy is still much lower than 53%. Moreover, the robust accuracy for PatchCleanser and a 20% patch in Table 1 is higher than that for a 15% patch (a *smaller* patch!). These results look suspicious and question the correctness of the Table 1 evaluation.
2. The experiment only uses 2000 images that are correctly classified and successfully attacked. This evaluation setup could introduce bias. For example, it misses the information for clean defense performance on images that are hard to attack.


B. The analysis of adaptive attack, which I believe should be one of the most important components of evaluation, is poorly written and unconvincing -- The paper simply claims that Corner Cutter “works” for adaptive attack but does not discuss *why* it works. Some experiment details are missing as well.
1. From my perspective, Corner Cutter shares a lot of similarities with earlier defenses like LGS and Digital Watermarking (DW). They all use some heuristics to detect and mask/smooth out the patch region: DW uses a saliency map; LGS uses pixel gradient; Corner Cutter uses frequency. DW and LGS are shown to be vulnerable to adaptive attacks by Chiang et al.. I don’t see what makes Corner Cutter so special to withstand adaptive attacks.

Chiang et al. https://arxiv.org/abs/2003.06693

DW https://openaccess.thecvf.com/content_cvpr_2018_workshops/papers/w32/Hayes_On_Visible_Adversarial_CVPR_2018_paper.pdf

LGS https://arxiv.org/abs/1807.01216

2. For the first adaptive attack.
    1. What does the accuracy of LaVAN in Table 3 mean?
    2. LaVAN is a targeted attack. I don’t think it is convincing to use a targeted attack success rate to evaluate the robustness against adaptive attacks. Instead, the attack success rate for untargeted attacks is a better metric.
    3. It is unreasonable to report averaged numbers for thresholds from 0.01 to 0.24 on a 0.01 increment. 0.01-0.24 is a wide range of thresholds. A lot of information is lost if the paper only reports an average. For example, some thresholds may work well for the adaptive attacks.
3. For the second adaptive attack for object detection. I don’t see why Corner Cutter can defend against naturalistic patches. Corner Cutter only leverages the pixel information to detect regions with high pixel frequency. This intuitively works for patches that look noisy and random. However, a naturalistic patch should look similar to a benign object; why would it introduce a high-frequency pixel pattern and be detected by Corner Cutter? I suspect that the authors did not implement the naturalistic patch attack properly. For example, the patch examples of Figure 14 all have high-texture patterns and look much less naturalistic to the examples in its original paper of naturalistic patch attacks.

C. Missing important citations. Just to name a few below.
1. https://openaccess.thecvf.com/content_cvpr_2018_workshops/papers/w32/Hayes_On_Visible_Adversarial_CVPR_2018_paper.pdf This paper also has a similar high-level pipeline of detecting patch region, masking, and inpainting.
2. https://arxiv.org/abs/2207.01795. This paper also uses a detect-and-mask defense strategy.
3. https://arxiv.org/abs/2111.09999 https://dl.acm.org/doi/abs/10.1145/3474085.3475653 two more papers study naturalistic patches, which I suspect could break Corner Cutter.

**Questions:**

1. could you explain the issues of Table 1 discussed in the weakness section?
2. could you provide more evidence for the robustness against adaptive attacks, as discussed in the weakness sections?

---

### Official Review · Reviewer_DrYS · 2023-10-30

**Soundness:** 2 fair
**Presentation:** 2 fair
**Contribution:** 2 fair
**Rating:** 3
**Confidence:** 5

**Summary:**

The paper proposes an approach to detect and remove the adversarial patches. For that, the images are transformed into a Fourier domain, and frequency values are centered, later, high-frequency discrepancies help in finding the adversarial patches. The experiments are performed using various adversarial patches including LaVAN and naturalistic patches.

**Strengths:**

* The paper is easy to read and follow.
* The proposed defense is straightforward and shows efficiency in detecting a variety of adversarial patches.

**Weaknesses:**

* The paper has not discussed or provided any insight on the defense which can simply fine-tune a classifier for patch detection. For example:

[1] Ojaswee O, Agarwal A, Ratha N. Benchmarking Image Classifiers for Physical Out-of-Distribution Examples Detection. In Proceedings of the IEEE/CVF International Conference on Computer Vision 2023 (pp. 4427-4435).
* Comparison has been done with old works majority of them published before 2019. Further, the older attacks are used.
* The experimental setting is weak and has been performed using a single network.
* An ablation study concerning different patch types can also be added.
* In comparison to the success in Table 1, Table 2 shows the drastic drop in the mAP which reflects the significant limitation of the proposed work. While the values are better than existing works, they are significantly low, especially under the 20\% pixels (30\% object) setting. The prime concern with this is that it makes the proposed defense "another" defense in the literature.
* Further, the proposed defense is vulnerable under an adaptive attack setting and seems easy to break. Interestingly, under the 20\% object setting, the defense shows a slightly better value than the non-adaptive attack, any observations?

**Questions:**

Please check the weakness section.

---

### Official Review · Reviewer_r6Dg · 2023-10-31

**Soundness:** 2 fair
**Presentation:** 3 good
**Contribution:** 2 fair
**Rating:** 3
**Confidence:** 4

**Summary:**

The authors propose CornerCutter, a novel empirical defence against adversarial patch attacks based on localizing, deleting and inpainting the patch in the image. They use frequency domain to localize the patch. CornerCutter is evaluated for the image classification and object detection tasks

**Strengths:**

Originality. The paper considers an empirical defence against adversarial patches that uses analysis in frequency domain. As authors state, this idea was used for classical noise-based perturbations but not studied well-enough for localized attacks (up to my knowledge).

Quality. The authors evaluate their method method on a wide range of datasets and compare to a wide range of defence methods. They also make an attempt to evaluate CornerCutter against adaptive attacks (Section 5.2) which is paramount for empirical defences.

Clarity. The paper is well-written and easy to follow. The experimental setup is discussed explicitly (Section 5).

Significance. As the authors state in the introduction, the safety of object detection algorithms is pivotal for many real-life applications and defending against patches is an important factor of it.

**Weaknesses:**

1. Quality. I have some concerns about your evaluation protocol. Why do you compare your method to PGD on Xview dataset and to Naturalistic patch on INRIA person dataset? Why not evaluate both methods on the same dataset or on both? I understand that that would require additional computational resources but proper evaluation of empirical defences has to be rather thorough, otherwise what you show could be just lucky cases.

2. Clarity. If you still decide to compare CornerCutter to different methods on different datasets, please state it more clearly in the experimental results e. g. in Table 2. From the caption and header it is not clear that different datasets are used. Therefore, very different benign accuracy for PGD and Naturalistic Patch is misleading since the attack should not affect benign accuracy.

3. Quality. Section 5.2 doesn’t sound convincing although evaluation against adaptive attacks is crucial for presenting empirical defences. You use different adaptive strategies: frequency clipping for image classification vs including frequency magnitude in the loss for object detection. Therefore, you conclusion about CornerCutter working well against adaptive attacks on object detection sounds a bit unjustified. What if you use frequency clipping as you did for the LaVAN patch? Also I would include more diverse attacks there including the ones not based on gradient descent, for example, Sparse-RS [1].

4. Quality. I did not find many details on the inpainting part although it seems to be one of the crucial steps of the method. In particular, what dataset was it trained on? Have you checked whether there are any data intersection with the datasets on which you evaluate? Access to an additional dataset for training the inpainting model is a strong assumption of the CornerCutter defence that gives it advantage over other baseline methods.

5. Clarity. Not necessarily a weakness, but it is a bit hard to navigate the paper as a whole since there are very few references to appendix from the main paper. For example, pointing the reader to Appendix B in Section 5 would help to navigate the paper in its fullness.

[1] Francesco Croce, Maksym Andriushchenko, Naman D. Singh, Nicolas Flammarion, and Matthias Hein. Sparse-RS: a versatile framework for query-efficient sparse black-box adversarial attack, 2020

**Questions:**

1. Have you considered computational overhead of your method? I would assume that inpainting may be a time consuming part. That would limit the defence application in real-time tasks such as autonomous driving.

2. There seem to be rather strong fluctuations in benign accuracy for the same method e. g. for JPEG it is 79.1, 83.5, 85.0, 79.4, … Do you have an explanation for that?

3. Could you formally specify, how exactly the phase shift mentioned in Section 4.1 is performed?